# Teachers' Attitudes toward Educational Inclusion in Spain: A Systematic Review

**Irene Lacruz-Pérez** *[ID]**, Pilar Sanz-Cervera** [ID] **and Raúl Tárraga-Mínguez** [ID]

Department of Education and School Management, Faculty of Teacher Training, University of Valencia, 46022 Valencia, Spain; pilar.sanz-cervera@uv.es (P.S.-C.); raul.tarraga@uv.es (R.T.-M.)
* Correspondence: irene.lacruz@uv.es

**Abstract:** Inclusive education is currently one of the main aspirations of the Spanish educational system and one of the key aspects for its achievement is teachers' attitudes toward educational inclusion. In recent years, many studies worldwide have analyzed this aspect, but so far, any systematic review has specifically focused on the Spanish educational framework. For this reason, the purpose of this paper is to review the studies published from 2010 to 2019 whose aim was to analyze teachers' attitudes towards educational inclusion in Spain. After a literature search in four different databases (PsycInfo, ERIC, Dialnet Plus, and Google Scholar), 34 studies were selected and reviewed. The results suggest that Spanish teachers' attitudes toward educational inclusion are generally positive, although in some cases they are ambiguous. Teachers' attitudes are mainly influenced by the amount of training and their contact or not with students with special educational needs. The discussion highlights that more studies with a greater methodological diversity are required in order to provide a complete analysis of teachers' attitudes toward inclusion and that teacher training is one of the best tools to generate positive attitudes.

**Keywords:** attitudes; inclusive education; systematic review; teachers

## 1. Introduction

Inclusive education seeks to achieve the highest levels of presence, participation and learning of all students in the regular educational system, especially of those who are in a vulnerable situation [1]. This idea has been supported by different international regulations such as the Salamanca Statement [2], the Convention on the Rights of Persons with Disabilities (CRPD) [3], or the 2030 Agenda for Sustainable Development [4].

In particular, after the approval of the CRPD [3] different countries began to reform their national legislation to bring their educational systems closer to the inclusion paradigm [5]. Nevertheless, there is some controversy around the meaning of the term "educational inclusion", since article 24 of the CRPD does not define it explicitly [6]. This lack of precision in its definition has generated some debate about the meaning of educational inclusion, since it has been understood as incompatible with special education [7], a schooling form that is considered essential to teach some students with a moderate or high degree of disability [8].

Therefore, inclusive education generates a substantive debate that should not be avoided, but rather approached with honesty and in defense of the interests of students and their families. This debate arises from the complexity of the subject, and it is also frequently distorted by the fact that there are numerous cases in which schools are required to implement inclusive practices without providing them with sufficient human, material and organizational resources to do so. This situation means that, probably, the most prudent attitude toward inclusion is to encourage (but not demand) the implementation of inclusive policies and practices.

Regarding the Spanish educational system, the idea of inclusive education established by national educational legislation does not correspond to what some authors qualify as

"full inclusion", understood as the schooling of every student in regular schools, regardless of his or her special educational needs [6,8]. On the contrary, even though the national education laws promote the schooling of students with special educational needs in regular schools since 1990 [9–12], they also support the schooling of some students in special education schools when the measures of attention to diversity that they require cannot be provided in regular schools.

In Spain, there are three main different types of schooling: regular schools, special education schools, or special education classrooms located in regular schools. In the last case, students with disabilities attend some hours to the regular classroom and spend the rest of the school time with the special education teacher in another classroom. These three major forms of schooling, which include different resources and supports, are the embodiment of the national educational policy regarding inclusion. Moreover, they are implicitly the recognition of the existence of a wide range of diversity among students, a diversity that also requires diversity in the educational response.

The decision of which type of schooling the students follow is made after the individual analysis of each case. The characteristics of each child, his or her diagnosis, and the degree of severity are taken into account when making the decision on the type of schooling. During the decision process, it is mandatory to listen to the assessment of the families themselves about the decision of the type of schooling of their children. Furthermore, the decision on the modality of schooling is evaluated (and if necessary, modified) after finishing each school year.

The last Spanish educational law approved in 2020 [12] has established that in a period of ten years it is intended to provide regular schools with more and better resources to teach students with disabilities, but special education schools will continue to be financed, since it is not possible always to enroll students with special educational needs in regular classrooms. The responsibility of putting these educational policies into practice on a day-to-day basis lies with the teachers, who are one of the main responsible for providing an adequate response to diversity. For this reason, their attitudes toward inclusion are a cornerstone in order to materialize the above-mentioned legislation into real inclusive practices. The study of teachers' attitudes toward inclusion allows us drawing up a profile about teachers' conceptions and at the same time, it lets us to know how to work to improve them [13]. Given its importance, this research area has progressively increased in recent years [14], being currently of great interest to the scientific community.

International research on this topic has been synthesized in several systematic reviews that reveal neutral attitudes towards educational inclusion [15–17], probably more consistent with the concept of integration than with the idea of inclusion.

Reviews focused specifically on the physical education area have even found negative attitudes towards inclusion [18]. In these studies, some specialists of this field give support to striking arguments as that inclusion is detrimental to the performance of students without special educational needs. In these researches, three groups of variables that may influence teachers' attitudes towards inclusive education have been identified.

First, in relation to the type of the students' diagnosis, teachers' attitudes tend to be more positive towards students with sensory or physical functional diversity than towards students with cognitive functional diversity or behavioral problems [15–17].

Second, regarding teachers' personal characteristics, it has been found that having prior experience in inclusive practices and having received training in special education is related to better attitudes towards inclusion [15–18]. Some studies have also found that younger and less experienced teachers are more open to inclusive education [15,17], although other studies have obtained the opposite [18].

Third, considering the educational environment, some studies have found that having enough material and human resources, as well as having the reinforcement of the school management team, influences the attitudes of teachers towards inclusion [15]. Additionally, teachers tend to consider the difficulty of the academic content incompatible with

inclusion, so they show less compliance with inclusive education in higher educational stages [15,18].

In recent years, several studies aimed at analyzing Spanish teachers' attitudes toward inclusion. Nonetheless, no systematic review has brought together the conclusions of these studies carried out within the framework of the Spanish educational system in a single study so far.

Therefore, the aim of this paper is to review the studies published in the last decade (2010–2019) that have analyzed the attitudes of teachers toward educational inclusion in any region of the country. Specifically, this review intends to answer the following questions:

1. How are the attitudes of pre-service teachers and in-service teachers toward inclusion in Spain?
2. Which research design has been followed in the different studies reviewed?
3. Which factors are related to teachers' attitudes toward educational inclusion?

## 2. Materials and Methods

A literature search was carried out in PsycInfo, ERIC, Dialnet Plus, and Google Scholar databases using the keywords *attitudes* and *teachers* combined with *inclusion*, *integration*, *inclusive education*, *disability*, or *diversity* (in English and Spanish).

Dialnet Plus includes almost 11,000 scientific journals, being probably the most exhaustive database in terms of publications at a national level. This fact makes it one of the reference databases in Spain, which is important to this review taking into account that it is limited to Spanish territory. The searches in this database were limited to studies conducted in Spain and the keywords were entered in any field.

PsycInfo and ERIC are two of the most widely used databases in specialized bibliographic searches in Psychology and Education. They were mainly used to identify articles published in international journals that might not be indexed in Dialnet Plus. In these two databases the keywords were searched in any field of the article except full text and they were limited to studies carried out in Spain.

Google Scholar is probably one of the databases that offers the highest amount of results, since it indexes publications of very different types. This database was used to identify publications that previous databases had not been able to locate. In this case, the keywords were entered in the search field, and the results were ordered by relevance. In the search carried out, the titles of the first 500 results were reviewed, since, due to the very nature of the database, the amount of results was practically unmanageable.

In all cases, searches were limited to works published from 2010 to 2019. The following criteria were taken into account in the selection of the articles: (a) studies that analyzed the perception or attitudes of teachers towards inclusive education; (b) the sample was pre-service or in-service teachers of early childhood, primary or secondary education; (c) quantitative or qualitative methodology was used; (d) works published in peer-reviewed scientific journals; (e) articles written in Spanish or English; and (f) conducted in Spain.

Figure 1 summarizes the search process that concluded with the selection of 34 articles.

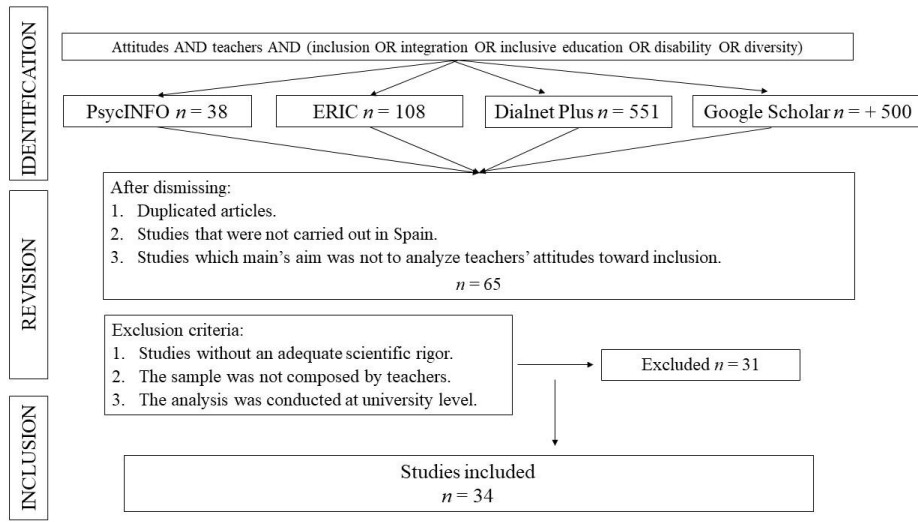

**Figure 1.** Search process flowchart.

### 3. Results

The results of the review of the 34 studies are distributed in five different tables according to the type of sample. Table 1 includes the results of 14 studies of which the sample were in-service teachers. Table 2 summarizes the studies whose participants were preservice teachers. The studies that analyze pre-service and in-service teachers' attitudes are included in Table 3. Table 4 contains the research focused just on special education teachers, and Table 5 refers to the studies carried out only with physical education teachers.

The total sample includes 7158 teachers (5372 in-service and 1786 pre-service), including early childhood, primary and secondary education stages. All the studies were carried out in Spain, and Andalusia is the region with the highest representation (12 studies). The research tools used in the different studies are diverse, although the most widely used is the *Scale of Attitudes towards People with Disabilities* [19]. It is used in eight of the works, which are mainly focused on pre-service teachers (see Table 2).

In 21 of the studies the conclusions indicated that the attitudes of teachers towards inclusive education are positive; in ten studies it is concluded that the results are mixed; and only in three studies results pointed out unfavorable attitudes. In total, considering the different studies, the attitudes of teachers have been analysed in relation to up to twenty variables. Mainly five of these variables stand out: (1) gender; (2) teaching experience; (3) contact with people with functional diversity and/or students with special educational needs; (4) the educational stage (early childhood, primary or secondary education); and (5) the amount of teachers' training in the inclusive education field.

Of the eighteen studies that analyzed the influence of gender, seven found that women tend to be more positive [20–26] and four found that men have better attitudes [27–30].

Regarding teaching experience, 10 studies analyzed its influence (see Tables 1 and 4): three found that teaching experience has a positive correlation with positive attitudes toward inclusion [27,28,31], three found the opposite [26,29,32], and the rest did not find a statistically significant relation [33–36].

Contact with people with disabilities and/or students with special educational needs seems to positively influence teachers' perception of inclusion, according to eight of the eleven studies that analyzed this aspect [20,23,24,31,37–40].

Concerning the educational stage, seven studies detected less favorable attitudes toward inclusion in secondary education than in primary education teachers [25,29,30,33–35,41]. At the same time, early childhood teachers' attitudes seem to be more positive than primary education teachers' attitudes [25,33,35,42].

A last important factor is training: in four studies with pre-service teachers (see Table 2), more positive attitudes were found in the final phases of their initial training compared

to the first courses of their training [20,40,42,43], and two other studies concluded that teachers who feel better trained are more positive towards inclusion [31,39].

**Table 1.** Review of the studies carried out with in-service teachers.

| Studies | Participants | Research Tool | Results |
|---|---|---|---|
| [44] | *n* = 143 (secondary education teachers). Years of experience: <5 (31%), 5–10 (25%), >10 (44%). Regular schools (86%), Semi-private schools (14%). | Questionnaire based on APADESO scale [45]. | Mixed attitudes. Teachers recognize the universal right to secondary education, considering that attention to diversity in mainstream classrooms is needed. However, they consider that integration is not very positive for their job: it lowers academic content, it impoverishes education, and capable students are neglected. |
| [39] | *n* = 36 (*n* = 2 early childhood, *n* = 12 primary, *n* = 22 secondary education). 69.4% women; 30.55% men. Age: 41–50 (56%). | Questionnaire developed and validated by the authors: Scale for Measuring the Attitude of the Regular Classroom teacher towards Educational Integration. Semi-structured interview. | Positive attitudes toward integration-inclusion. Having training, satisfactory prior experience and consistent expectations are related to better attitudes. Regarding the inclusion of students with autism (high performance), the support of the administration and the school environment influence teachers' attitudes (in secondary schools). |
| [30] | *n* = 77 (*n* = 32 primary; *n* = 45 secondary education) + *n* = 39 university. 57.25% women; 32.8% men. Age: 35–45 (42.2%). Experience with students with disabilities: 100%. Three regular schools. | Questionnaire developed and validated by the author. | They feel uneasy working with students with disabilities (especially women and secondary school teachers). |
| [32] | *n* = 336 (20.2% early childhood; 39.6% primary; 40.2% secondary education) 67% women; 33% men. Age: M = 41.5. Years of experience: <4–8 (28%), 9–15 (20.6%), >15 (51.3%). Experience with students with special educational needs: 100%. | The Teachers' Perception on Inclusion Questionnaire [46]. | Teachers perceive inclusion positively: they consider that it develops tolerance (84%), and it is unfair to separate students with special educational needs from the rest of the students (59%). Nonetheless, they also consider that inclusion is impossible for students with moderate-severe difficulties (60%), especially in secondary education (70%). Early childhood education teachers' have better attitudes toward inclusion than primary and secondary education teachers. Having support is related to better attitudes. |
| [31] | *n* = 20 (primary education). 40% women; 60% men. Age: >40 (70%) Years of experience: <5 (20%); 5–20 (30%); >20 (50%). Two regular schools. | Adaptation: [46] questionnaire. | Inclusion generates tolerant attitudes (90%), being possible in secondary education. It favors the teaching–learning process (75%), schooling of students with severe disabilities should be in regular schools (50%), they value support teachers positively (90%). Having training, experience, and contact with people with disabilities are related to better attitudes. |

**Table 1.** *Cont.*

| Studies | Participants | Research Tool | Results |
|---|---|---|---|
| [34] | *n* = 82 (early childhood, primary and secondary education) 72% women; 28% men. Age: M = 39.51. Years of experience: M = 1.46. Experience with students with special educational needs: 100%. | Teachers' attitudes and practices concerning inclusion [47]. | Positive attitudes. Inclusion encourages tolerance. Teachers of regular and semi-private schools, who teach in early childhood or primary education show better attitudes. |
| [35] | Questionnaire: *n* = 2518 teachers (20.8% early childhood, 56% primary, 23.2% secondary education). Years of experience: 0–9 (54.8%); 10–20 (29.2%); >20 (16%). Interview: *n* = 27 teachers. | Questionnaire specifically elaborated for this study. Semi-structured interview, developed and validated by authors. | Inclusion implies multiple advantages: affective and socio-emotional development of students with special educational needs; students without special educational needs acquire ethical values; greater involvement of families and social agents; and acceptance of society. However, inclusion harms the academic performance of all students. Early childhood education teachers perceive inclusion significantly better, followed by primary education teachers and finally secondary education teachers. More experienced teachers perceive more advantages for students with special educational needs and less experienced teachers perceive more benefits of inclusion for students without special educational needs. |
| [48] | *n* = 46 teachers (early childhood and primary education). A regular public school with students with hearing impairment and a regular private school. | Adaptation: Questionnaire of Opinions, attitudes and competences of teachers towards disability [49]. | Teachers positively value teacher cooperation and their awareness toward disability (it seems to be better in the public school). |
| [26] | *n* = 407 (secondary education). 47.4% women; 52.6% men. Age: 31–40 (41.9%). Years of experience: 4–6 (29.2%) Inclusive education training: 39.8%. | Questionnaire developed and validated by the authors. | Mixed attitudes. Most of the secondary teachers consider that attention to diversity is a duty of the school (90%); inclusion is important (72.6%); it enriches the school community (68.7%); the education of the students with special educational needs is the responsibility of both the regular teachers and the special education specialists (76.9%). However, reaching real inclusion is utopian (46.8%); inclusion implies extra work to teachers (78.7%); students with disabilities should be educated in special education schools (44%). Having more teaching experience is related to less favorable attitudes. |
| [50] | *n* = 7 (early childhood and primary education teachers). Years of experience in inclusion: minimum 5. | Questionnaire (open-ended questions). | Teachers accord importance to inclusion (it also benefits students without special educational needs), although their discourse is integrator and not inclusive. |
| [36] | *n* = 78 (early childhood education teachers). 91% women; 9% men. Age: 21–30 (21.8%); 31–40 (25.6%); 41–50 (33.3%); 51–60 (19.2%). | Scale of Attitudes towards People with Disabilities [19]. | Early childhood education teachers recognize people with disabilities rights' and the importance of their social inclusion. Teachers from the first years of early childhood education (0–3) have less positive attitudes than teachers from the last years of this stage (3–6). |

**Table 1.** *Cont.*

| Studies | Participants | Research Tool | Results |
|---|---|---|---|
| [29] | *n* = 175 (10.9% early childhood, 36% primary, 53.10% secondary education teachers). 70.9% women; 29.1% men. Age: M = 40.13. Contact with students with disabilities: 100%. Urban area (74.9%), rural (25.15%). | Adaptation: questionnaire for teachers about attitudes towards students with special educational needs derived from disability [51]. | Teachers accord greater importance to socio-emotional aspects of students with disabilities than to academic aspects. Teachers from semi-private schools feel more trained for inclusion. Having better attitudes is related to rural areas, to men (not significantly), to have less teaching experience and to early childhood and primary education teachers. |
| [28] | *n* = 402 (48.9% early childhood education, 15% primary education, 5.1% special education, 13.8% other, 16.7% secondary education). 63.7% women; 36.3% men. Years of experience: <5 (26.5%); 6–15 (28%); 16–29(25.1%); >30 (18.2%). | Questionnaire specifically elaborated for this study. | Teachers show positive attitudes toward diversity, although they do not know how to organize the educational response. Men and more experienced teachers value inclusive polices better. Special education specialists value inclusive culture and practices significantly better. Teachers from regular schools are more positive towards inclusion. |
| [52] | *n* = 30 (*n* = 6 early-childhood education; *n* = 18 primary education; *n* = 2 physical education; *n* = 2 English; *n* = 1 religion; *n* = 1 music). | Scale [30]. | Teachers feel uneasy with students with disabilities. They consider that special education specialists are the responsible of teaching these students. |

**Table 2.** Review of the studies carried out with pre-service teachers.

| Studies | Participants | Research Tool | Results |
|---|---|---|---|
| [42] | *n* = 274 pre-service teachers (21.8% early childhood; 72.4% primary education). 91.5% women; 9.4% men. Age: M = 22.16. Grade: 2nd (*n* = 112), 3rd (*n* = 44), Special Education Master's Degree (*n* = 59). | Adaptation: Opinions Relative to Integration of Students with Disabilities Scale [53]. | Positive attitudes. Master's degree students show better attitudes than early childhood pre-service teachers; and both of them have better attitudes than primary education pre-service teachers. Training in special education programs improves the attitudes of second-grade students. |
| [24] | *n* = 91 pre-service teachers (early childhood and primary education) 78% women; 22% men. Age: M = 28.13. Contact with students with special educational needs: 74.7%. | Questionnaire developed and validated by the authors. | Women and those who have had contact with students with special educational needs feel more prepared for inclusion. Nonlinear relation of teachers' attitudes with contact frequency was found. |
| [20] | *n* = 315 pre-service teachers (early childhood and primary education), and educational psychologists). Grade (early childhood): 1st (*n* = 43); 3rd (*n* = 39) Grade (primary) 1st (*n* = 36), 3rd (*n* = 37). Grade (educational psychologists) 1st (*n* = 82), 2nd (*n* = 76). 265 women; 50 men. Age: M = 22.35. Experience with students with special educational needs: 43.20%. | Adaptation: Attitudes toward Inclusive Education [54]. Reduced version: Values Questionnaire [55]. | Positive attitudes: concern for equality; willingness to make curriculum more flexible and to modify spaces. Women pre-service teachers, older participants, and participants with prior experience with students with special educational needs show slightly more positive attitudes. Third-grade pre-service teachers' show better attitudes than First-grade pre-service teachers (non-linear relation). |

**Table 2.** *Cont.*

| Studies | Participants | Research Tool | Results |
|---|---|---|---|
| [56] | *n* = 99 pre-service teachers (secondary education). 60.6% women; 39.4% men. Age: <25 (63.6%). | Questionnaire developed and validated by the authors. | Favorable attitudes. Pre-service teachers consider attention to diversity as an enrichment and duty of schools. They agree with combined schooling and they think inclusion is possible in secondary education. They consider that teaching students with special educational needs is everyone's responsibility, although it involves extra work to the regular teachers. |
| [38] | *n* = 41 pre-service teachers (early childhood education) 92.7% women; 7.3% men. Age: M = 21.37. University access: Entrance examination (53.7%); Higher Level Education Cycle (46.3%). Contact with people with disabilities: 39%. | Scale of Attitudes towards People with Disabilities [19]. | Positive attitudes. Early childhood pre-service teachers recognize the rights of people with disabilities and they intend to interact with them. They value their abilities less positively. Pre-service teachers who have accessed university through entrance examination and have had prior contact with people with disabilities show better attitudes towards inclusion (no strong relation). |
| [57] | *n* = 65 pre-service teachers (primary education) 38 women; 27 men. Age: M = 21.28. | Scale of Attitudes towards People with Disabilities [19]. | Positive attitudes, but not enough (especially concerning pre-service teachers' expectations towards students with disabilities and their knowledge about these students). |
| [23] | *n* = 107 pre-service secondary education teachers (psychologic counselling *n* = 12; Mathematics *n* = 13; Spanish language *n* = 33; Geography and History *n* = 22; Physical education *n* = 11; Others *n* = 16). 57% women; 43% men. Age: M = 26.14. Contact with people with disabilities: 62.6%. | Scale of Attitudes towards People with Disabilities [19]. | Positive attitudes. Women and those who have had contact with people with disabilities score better (not significantly). Geography and History specialists show less favorable attitudes. |
| [43] | *n* = 158 pre-service teachers (early childhood and primary education). Grade: 1st (*n* = 90), 4th (*n* = 68). 76.8% women; 23.2% men. Age: 18–22 (77.3%). | Adaptation: [58] questionnaire (Likert items and open-ended questions). | Positive attitudes. Pre-service teachers consider that attention to diversity is important; it enriches the school community; it allows teaching all students fairly; and it promotes positive values. However, they are undecided about the best type of schooling for students with special educational needs and the extra work that the inclusion of these students in the regular classroom implies for the teacher. Fourth-grade pre-service teachers have significantly better attitudes and richer and more realistic speeches than first-grade pre-service teachers. |
| [37] | *n* = 48 pre-service teachers (early childhood education). 91.67% women; 8.33% men. Age: M = 21.25. | Scale of Attitudes towards People with Disabilities [19]. | Pre-service teachers who have greater contact with people with disabilities have greater willingness to interact with them. Remarkable (not significant) relation between the attitudes of the participants and their academic performance was found. |

**Table 2.** *Cont.*

| Studies | Participants | Research Tool | Results |
|---------|-------------|---------------|---------|
| [41] | *n* = 46 pre-service teachers (early childhood and primary education) 78% women; 22% men. *n* = 50 pre-service secondary education teachers. 50% women; 50% men. Age: 22–26. | [58] questionnaire. | Positive attitudes. Pre-service teachers consider that inclusion is enrichment. In early childhood and primary education, pre-service teachers are undecided about the appropriate schooling modality of students with disabilities. They do not know sure if teaching students with special educational needs implies an extra work for the teachers. Secondary education pre-service teachers have less positive attitudes. |
| [25] | *n* = 314 pre-service teachers (13.1% early childhood, 19.1% primary, 34.1% secondary education, 33.8% other). 75.45% women; 24.52% men. Contact with people with disabilities: 55.8%. | Scale of Attitudes towards People with Disabilities [19]. | Pre-service teachers show positive attitudes. Women have significantly better attitudes than men. Early childhood pre-service teachers show more positive attitudes than primary education pre-service teachers, and they are also more positive than secondary pre-service teachers. |
| [59] | *n* = 120 pre-service teachers (early childhood education); *n* = 16 English specialists; *n* = 34 special education teachers; *n* = 139 physical education teachers; *n* = 11 creative languages; *n* = 46 without specialization 95.1% women; 4.8% men. Age: M = 22.39. | Scale of beliefs towards attention to disability in physical activity [60]. | Positive attitudes toward disability: especially special education pre-service specialists (not significantly). |

**Table 3.** Review of the studies carried out with in-service and pre-service teachers.

| Studies | Participants | Research Tool | Results |
|---------|-------------|---------------|---------|
| [21] | *n* = 26 (primary education) 17 women; 9 men. Age: M = 43.42 *n* = 26 pre-service teachers (primary education) 18 women; 8 men. Age: M = 27.73 Contact with people with disabilities: 100%. | Scale of Attitudes towards People with Disabilities [19]. | Positive attitudes toward disability (especially in women). |
| [61] | *n* = 4 pre-service teachers (early childhood and primary education) Grade: 1st (*n* = 2), 4th (*n* = 2). *n* = 2 in-service teachers (first year of teaching: early childhood and primary education). 3 women; 3 men. | Adapted interview from [62,63]. | Both pre-service and in-service teachers show inclusive attitudes. They consider that inclusion fosters cooperation, empathy, and tolerance, among other values. However, they relate the benefits of inclusion only to vulnerable students (integration approach). |

Table 4. Review of the studies carried out with special education teachers.

| Studies | Participants | Research Tool | Results |
|---|---|---|---|
| [32] | *n* = 106 special education specialists (primary and secondary education). 82.7% women; 17.3% men. Age: M = 37.8 Years of experience: M = 12.3 | Adaptation: Attitudes towards integration in primary [64] and secondary [65] education questionnaire. | They consider integration does not work properly. Lower acceptance of students with behavioral problems and belonging to ethnic minorities. Having more experience and being older are two aspects associated to less positive teachers' attitudes. |
| [66] | *n* = 428 teachers of special education schools (80.3% special education teachers; 9.6% speech-language teachers; 10.1% others) 73.1% women; 26.9% men. Years of experience: <1–7 (41.4%); 8–14 (12.7%); >15 (45.8%). | Questionnaire on training needs of teachers in special education schools, developed and validated by authors. | Teachers consider inclusion a basic pillar of education. Special education schools should be based on the inclusion principles. Teachers without experience perceive inclusion more favorably. |
| [27] | *n* = 428 special education specialists. 73.1% women; 26.9% men. Age: >41 (49.7%). Years of experience: <1–3 (27.3%); 4–7 (14.5%); 8–14 (14.2%); >15 (44%). | Questionnaire developed and validated by the authors. | Special education specialists consider diversity important. They favorably perceive inclusive practices in special education schools. Having less teaching experience is related to less positive attitudes. Men seem to have better expectations towards students with disabilities than women. |

Table 5. Review of the studies carried out with physical education teachers.

| Studies | Participants | Research Tool | Results |
|---|---|---|---|
| [67] | *n* = 7 physical education specialists (*n* = 2 pre-service teachers; *n* = 2 primary education in-service teachers; *n* = 3 secondary education in-service teachers) Years of experience: primary 5–10; secondary 4–8. | Open-ended questions posed by authors. Two sessions' observation. | Positive attitudes. Except in cases of students with very specific needs, teachers generally agree with schooling in regular schools. |
| [40] | *n* = 76 physical education pre-service teachers. 34 women; 42 men. Age: M = 22.61. Grade: 3rd (*n* = 42); 4th (*n* = 34). Years of experience with people with disabilities: <1 (75%); 1–3 (14.5%); 3–5 (3.9%); >5 (1.3%); No answer (5.3%). | Scale of Attitudes towards People with Disabilities [19]. Scale of beliefs towards attention to disability in physical activity [60]. | Very positive attitudes toward disability were found. Having prior experience with people with disabilities is related to better attitudes (not significantly). |
| [22] | *n* = 40 physical education specialists. 32.5% women; 67.5% men. Age: <30 (*n* = 9); 31–40 (*n* = 22); >41 (*n* = 9). | Adaptation: [68] questionnaire. | Not very positive attitudes were found. Some teachers consider that it is impossible to work with the rest of students effectively (37.5%); "handicapped" students should study in special education schools (35%). Middle-aged men have moderately less positive attitudes. |

## 4. Discussion

This review lets us to systematize in a single study the conclusions obtained on teachers' attitudes toward inclusive education studies carried out in Spain in the last decade, and it enables us to answer mainly three questions.

The first question refers to know how Spanish pre-service and in-service teachers' attitudes towards inclusive education are. Regarding the studies focused just on in-service teachers, seven studies out of 14 found positive attitudes. The conclusions of nine of the

11 studies conducted with pre-service teachers also show positive attitudes. The analysis carried out with special education teachers show positive attitudes in two out of three, and the same happens in the studies focused on physical education teachers. Therefore, considering the results obtained in most of the studies, teachers' attitudes toward inclusion tend to be positive.

Nevertheless, in ten studies the attitudes are mixed: five of them are in-service teachers' studies [26,33,35,44,50], three are pre-service teachers' research [24,37,57], another one is a mixed study with in-service and preservice teachers [61], and the last one that found mixed attitudes was conducted just with special education teachers [32]. In three other cases, teachers' attitudes toward inclusion tended to be even negative [22,30,52].

Although the universal right to inclusion is generally recognized, sometimes teachers have some beliefs that limit their positive attitudes or that place them in an integrative approach rather than an inclusive approach. For instance, some teachers attach inclusion to the type of special educational needs and other teachers consider that inclusive education is detrimental to students without special educational needs. These results are in line with previous reviews [15–17]. This is an unfavorable result for inclusion since teachers play a central role in schools, and therefore they are the main ones in charge of transferring the regulations on inclusion to the classroom. In addition to this, the results that point to positive attitudes should be understood with caution as they may be influenced by the social desirability bias. Currently, inclusive education is a sensitive issue, which has occasionally starred in debates with relatively opposed positions, so it is possible that teachers respond to it according to what is socially acceptable [69].

This leads us to answer the second question, referring to the research design used in each study. Most of the reviewed studies (27 out of 34) use questionnaires with a Likert response as a research tool. Some authors explain that questionnaires only allow us to know explicit attitudes. Consequently, they suggest expanding the study to implicit attitudes, since they seem to be less susceptible to social desirability bias [69]. For this reason, research tools such as the Single-Target Implicit Association Test (ST-IAT) [70] could be used. ST-IAT is an open-source computer tool that is available to the research community to be replicated in other contexts. During its implementation, a series of stimuli (some of them related to inclusion) are displayed to participants. They must react to these stimuli by pressing two keys, choosing them according to whether they are words with positive or negative emotional valence. By measuring the latency of the different evaluation blocks that are carried out and doing the corresponding calculation, a score is obtained. This score allows researchers to read whether the implicit attitudes of the participants towards inclusion are positive or negative [69]. Qualitative studies should also be carried out through interviews, discussion groups or sessions observation (methodologies that are included only in four of the reviewed articles). The use of these methodologies would allow teachers to express their positions and beliefs about inclusion with more nuances, thus obtaining a more complete analysis of the school reality. In addition, studies that combine several methodologies are needed, since this would allow a triangulation of the results in order to be able to evaluate the coherence between the explicit statements of the teachers and their educational practice.

The third question to which this review can answer refers to the great variety of factors that can be related to teachers' attitudes towards inclusion. In most of the studies, twenty-seven specifically, variables related to some teachers' characteristics have been analyzed. In 13 studies some variables related to the school environment, and in just three studies some variables related to the students have also been analyzed. According to some international reviews [15–18], training in special and inclusive education, as well as contact with people with functional diversity or students with special educational needs positively influence teachers' attitudes towards inclusion. The results of this study also show, in line with [15] and [18], that there is a relationship between the educational stage and the teachers' attitudes, being attitudes in higher stages less favorable. Differences have also been found in the attitudes of teachers depending on their position as regular teachers or

special education specialists. Although this relation has only been analyzed in two studies, both have found better attitudes in special education teachers [28,42]. Additionally, other studies highlight some teachers' beliefs that go against inclusion, such as the fact that just special education specialists should be in charge of the education of students with functional diversity [22,26,52], or that inclusion involves "added" work to the regular teachers [26,43,56].

The relationship between these factors and teachers' attitudes toward inclusion leads us to highlight the importance of initial training, especially the one of regular and secondary education pre-service teachers, to generate positive attitudes. During their training, pre-service teachers should acquire knowledge, strategies, and skills, and they should learn about the available resources to teach students with special educational needs. Likewise, they should work on reflective practice to understand the meaning of inclusion and consider it part of their job and teaching responsibility, thus avoiding conceiving it as an extra work. It is also essential to give them opportunities to put these learnings into practice and to have direct contact with students with special educational needs.

These conclusions should be understood taking into account some study limitations. First, the studies included in this review are mostly quantitative, a methodology that does not allow describing in depth a concept with as many edges as the attitudes concept has. Second, the number of the studies that analyze the relation between teachers' attitudes and other variables related to the students' characteristics and contextual characteristics of the school environment is disproportionate compared to those that analyze teachers' attitudes and other variables related to the teachers' characteristics. This decompensation makes the comparison of the influence of these three factors on teachers' attitudes toward inclusion difficult. Therefore, it does not allow us to know exactly the type of variable that has the greatest influence on them. Finally, most of the studies reviewed have referred to attitudes toward special educational needs as a generic or global construct, ignoring the multiple nuances it may contain. Although the label of special educational needs is a very broad term that includes students with very different characteristics, practically none of the studies reviewed analyze whether teachers' attitudes toward inclusion are modulated by the type of special educational need of the students or other characteristics (such as their sex, nationality, religion, or social status). In other words, although there is diversity in diversities, very few studies have analyzed how this circumstance influences the teachers' attitudes.

Relying on these limitations, we propose as future lines of research: to increase triangulation in the analysis of teachers' attitudes toward inclusion; to conduct qualitative studies; to study in depth the relation between teachers' attitudes and some factors related to the students and the school environment characteristics; to analyze with greater precision the influence of training on teachers' attitudes; to analyze whether teachers' attitudes toward inclusion are modulated by the characteristics of the students; and to extend the study of attitudes toward inclusion to the entire school community (not only teachers).

## 5. Conclusions

The present systematic review allows us to draw at least three fundamental conclusions:

The first conclusion refers to the methodology used to date to study attitudes toward inclusion. Researchers interested in this field of study must take note that there is excessive homogeneity in the methodological approach used to analyze attitudes toward inclusive education.

As it has been evidenced in this review, and as it has also been found in other reviews carried out at an international level [17,71], the attitude questionnaires using Likert-type items are practically the only type of instruments that has been used in this area. The conclusion, therefore, is clear: if we continue to study attitudes only through Likert scales, we will be looking again and again at a complex reality only from one point of view, which provides us with a specific type of scales that can also be notably bound to social desirability bias [69,70,72].

If we truly aspire to be able to study the phenomenon of attitudes in all its complexity, it is necessary to complete the information provided by these scales with other instruments that allow us to contemplate attitudes from other angles, thus allowing us a triangulation of results. Conducting interviews, focus groups, observation records by professionals outside the schools and the application of instruments that evaluate not only explicit but also implicit attitudes, can be very useful ways that allow us to get closer to know about the teachers attitudes towards inclusive education in greater depth.

Second, the results of the review show us that teachers' attitudes towards inclusion are not decidedly positive. There are numerous nuances regarding teacher attitudes. There are differences between special education teachers and regular teachers; and there are differences between teachers at different stages of the educational system. Altogether, it seems that teachers' attitudes are closer to the idea of integration than to the idea of inclusion. This result is problematic since educational policies in Spain in recent decades tend towards increasingly inclusive models (thus surpassing the integrating models). Therefore, it seems that there is a certain unresolved discrepancy between what the legislation proposes and what a part of the teaching staff considers should be proposed. The debate on inclusive education is a legitimate and necessary debate, in which the different possible positions are welcome, if what they seek is to pursue the best results for students. However, we must be very cautious so that the outcome of this debate does not generate negative consequences in practice for the education of students.

Finally, the main conclusion that we can obtain from this review is that there is still a long way to go, a path that does not run through a single path, but through multiple paths in which it is worth moving forward. One of these paths is that of teacher training. Attitudes are not created from scratch. They are molded and shaped over the years from the experiences lived, and from the training received too. For this reason, teacher educators must take note that it is our responsibility to try to explain that the elimination of barriers to inclusion and the construction of more livable and more inclusive schools is positive for all those who live in the school and it is a must for students with special educational needs.

**Author Contributions:** Conceptualization, R.T.-M.; Methodology, R.T.-M. and I.L.-P.; Writing—original draft preparation, I.L.-P.; Writing—review and editing, R.T.-M., I.L.-P., and P.S.-C.; Project administration, P.S.-C.; Funding acquisition, P.S.-C. All authors have read and agreed to the published version of the manuscript.

**Funding:** This research was funded by Valencia Government grant number GV/2020/073. The APC was funded by Valencia Government. This study also received human and financial resources from the Valencian University, grant code UV-INV-PREDOC19F1–1010132.

**Informed Consent Statement:** Not applicable.

**Data Availability Statement:** No new data were created or analyzed in this study. Data sharing is not applicable to this article.

**Conflicts of Interest:** The authors declare no conflict of interest.

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
