# Peer review of "Teachers’ Attitudes toward Educational Inclusion in Spain: A Systematic Review"

_education, doi:10.3390/educsci11020058_

Round 1

Reviewer 1 Report

This is a very interesting systematic review of the teacher's attitude in the filed of mathematics in Spain. This work is timely and will be of help.

  • First of all, why did you choose that databases instead of other more related to area? Google Scholar is sometimes  confusing with the terms. Moreover, why didn't you use SCOPUS, WOS or evenmore MathSciNet?
  • Table 1 is really hard to follow, I suggest authors to break it into different smaller tables that allow reader to understand the point with it.
  • Conclussions Section needs a really deep improvement, it is just two paragrapahs sumarizing previous results please conclude.

Author Response

Dear Reviewer,

We are grateful for the contributions and suggestions, which we believe have notably improved the quality of the article. Below we expose the changes made in the article text (they are also marked on the document using the “Track Changes” function).

  1. First of all, why did you choose that databases instead of other more related to area? Google Scholar is sometimes confusing with the terms. Moreover, why didn't you use SCOPUS, WOS or evenmore MathSciNet?

The review is limited to studies carried out in Spanish territory exclusively. For this reason, it was not suitable to use the common international databases in this type of review.

Dialnet Plus was chosen because it is the referral database in Spain, since it includes practically all the scientific publications of this country.

Although Google Scholar has significant drawbacks, we also used it because it contains many Spanish publications that are not indexed in international databases such as Scopus, WoS, ERIC or PsycInfo.

ERIC and PsycInfo were chosen to identify studies conducted in Spain, but published in international journals specialized in education or psychology, since it was possible that these international journals were not indexed in Dialnet Plus.

Changes. Page 2, Section 2 “Materials and methods”:

  • We toggled the second and third paragraphs.
  • Line 93: we added the following sentence in order to clarify why we chose Dialnet Plus: “This fact makes it one of the reference databases in Spain, which is important to this review taking into account that it is limited to Spanish territory”.
  • Line 98: we added the following sentence to explain the reason why we used PsycInfo and ERIC databases: “They were mainly used to identify articles published in international journals that might not be indexed in Dialnet Plus”.
  1. Table 1 is really hard to follow, I suggest authors to break it into different smaller tables that allow reader to understand the point with it.

Changes. From page 4 to page 9:

In order to make the results table easier to read, we have divided Table 1 in five different tables. Some research included in this review analyse in-service teachers’ attitudes toward inclusion and other ones analyse pre-service teachers’ attitudes. Six studies focus on two types of teacher’s specialisation: physical education or special education. We have considered that the job or studying position of the participants is a relevant factor in the study of teachers’ attitudes toward inclusion and that is why we have clustered the articles according to it:

“Table 1. Review of the studies carried out with in-service teachers (Fourteen studies)

Table 2. Review of the studies carried out with pre-service teachers. (Twelve studies)

Table 3. Review of the studies carried out with in-service and pre-service teachers. (Two studies)

Table 4. Review of the studies carried out with special education teachers. (Three studies)

Table 5. Review of the studies carried out with physical education teachers. (Three studies)”.

Changes. The reorganisation of the results table has leaded us to make small changes at section 3 “Results” (pages 3 and 4) in order to give coherence to the text:

  • In the first paragraph (lines 136- 142) we have explained the division of the results in five different tables and we have pointed out which type of studies are included in each table.
  • Line 147: we have added the following sentence: “It is used in eight of the works, which are mainly focused on pre-service teachers (see Table 2)”.
  • Lines 160-161: we allude to Table 1 and Table 4.
  • Lines 171-172: we allude to Table 2.

Changes. For the same reason, we have also modified the second and the third paragraphs of section 4 “Discussion” (pages 9 and 10):

  • In the second paragraph (lines 192- 197), according to the studies division made in Tables 1,2,3,4 and 5, we have specified the number of studies from each table that have found positive attitudes toward inclusion.
  • In the third paragraph (lines 199- 203), we have done the same with the studies that found mixed attitudes toward inclusion.
  1. Section needs a really deep improvement; it is just two paragraphs summarizing previous results please conclude.

Changes. Conclusions section has been completely rewritten (page 12). We have extended this section to include the three main conclusions that could be drawn from this systematic review.

4. Some of the changes introduced in the text (such as restructuring the results table or adding new references) have modified the order of appearance of the references in the text. Consequently, we have changed the number of most of the references in the text and in the final reference list.

5. After reviewing the “Instructions for Authors” and rereading the full text, we have made minor changes (misprints, linguistic mistakes and format changes) in lines 13, 14, 15, 19, 38, 55, 56, 65, 67, 72, 158, 159, 163, 166, 209, 244, 253, as well as in Figure 1, Table 1 and the final reference list.

Reviewer 2 Report

The authors seem to assume that all inclusion is good. I think they should question that basic premise. Full inclusion--meaning each and every student with a disability should be in general education--is obviously unworkable, cruel, and irrational.

The authors need to explain the difference between integration and inclusion. They also need to explain what is meant by inclusion. It could mean participation in a special class in a regular school or participation in a regular class in a regular school. It could mean being in school but a special school or special class. 

Where to draw the line for inclusion is obviously a judgment call, but one every teacher must. Many children with disabilities can and should be included in general education, but children, their parents, and their teachers must judge what is best for a particular child.

The authors need to acknowledge this reality specifically rather than support the ideology of full inclusion with no regard for the characteristics and needs of students, teachers, and parents.

Attitudes toward inclusion could be positive in some cases and negative in others, depending on the particular disability. The authors also need to consider how diversity in ability is not like diversity in many other things, like color or social status or sex or religion or tribe or nationality. There is diversity in diversities, and all diversity can not be treated the same.

Author Response

Dear Reviewer,

We are grateful for the contributions and suggestions, which we believe have notably improved the quality of the article. Below we expose the changes made in the article text (they are also marked on the document using the “Track Changes” function).

  1. The authors need to explain the difference between integration and inclusion. They also need to explain what is meant by inclusion. It could mean participation in a special class in a regular school or participation in a regular class in a regular school. It could mean being in school but a special school or special class.

Changes. Page 1, section 1 “Introduction”:

After the first paragraph of the section, we have added two new paragraphs to explain the difference between “integration” and “inclusion” (lines 30-38). We have done it explaining the last educational laws in Spain, because they introduced both paradigms into the Spanish educational system.

We have also added a new sentence in the fourth paragraph of section 1 in order to give coherence to the text (lines 44-46). 

  1. The authors seem to assume that all inclusion is good. I think they should question that basic premise. Full inclusion--meaning each and every student with a disability should be in general education--is obviously unworkable, cruel, and irrational. Where to draw the line for inclusion is obviously a judgment call, but one every teacher must. Many children with disabilities can and should be included in general education, but children, their parents, and their teachers must judge what is best for a particular child. The authors need to acknowledge this reality specifically rather than support the ideology of full inclusion with no regard for the characteristics and needs of students, teachers, and parents.

Changes. Page 1, section 1 “Introduction”:

Lines 38-43: After explaining the difference between “integration” and “inclusion”, we have included a brief paragraph in which it is made explicit that although it is preferable for all children with special educational needs to be enrolled in regular classrooms, this is not always possible. The characteristics of each child, his or her diagnosis, and the degree of severity should be considered. It is the children themselves, their parents and teachers who should consider what is the best type of schooling in each case.

  1. Attitudes toward inclusion could be positive in some cases and negative in others, depending on the particular disability. The authors also need to consider how diversity in ability is not like diversity in many other things, like colour or social status or sex or religion or tribe or nationality. There is diversity in diversities, and all diversity cannot be treated the same.

This is an interesting idea in which we agree that it is necessary to analyse in depth. Unfortunately, this diversity in diversities has hardly been explored by any of the studies included in the review. For this reason, we have included the absence of this analysis in the study limitations (pages 16-17, lines 274-291).

4. Some of these changes introduced in the text (such as restructuring the results table or adding new references) have modified the order of appearance of the references in the text. Consequently, we have changed the number of most of the references in the text and in the final reference list.

5. After reviewing the “Instructions for Authors” and rereading the full text, we have made minor changes (misprints, linguistic mistakes and format changes) in lines 13, 14, 15, 19, 38, 55, 56, 65, 67, 72, 158, 159, 163, 166, 209, 244, 253, as well as in Figure 1, Table 1 and the final reference list.

Round 2

Reviewer 1 Report

Authors have addressed all my previous comments.

Author Response

Dear Reviewer,

Thanks a lot for your review. We consider that your proposals have considerably improved our work.

Best regards.

Reviewer 2 Report

This revision is an improvement over what was first submitted, but in my opinion it still needs substantially more improvement. The very concept of inclusion is not well discussed (e.g., does it mean actively engaged in learning appropriate skills or does it refer to bodily presence?). Furthermore, I feel the manuscript would be greatly improved by acknowledging the ways in which inclusion in Spain fits into international perspectives, including the United Nations' CRPD and other nations' initiatives. For example, the authors might well read and reference at least the following: 

Anastasiou, D., Gregory, M., & Kauffman, J. M. (2018). Commentary on Article 24 of the CRPD: The right to educationIn I. Bantekas, M. Stein, & D. Anastasiou (Eds.), Commentary on the UN Convention on the Rights of Persons with Disabilities (pp. 656-704). New York, NY: Oxford University Press.

Anastasiou, D., Felder, M., Correia, L. A. M., Shemanov, A., Zweers, I., & Ahrbeck, B. (2020). The impact of article 24 of the CRPD on special and inclusive education in Germany, Portugal, the Russian Federation, and the Netherlands. In J. M. Kauffman (Ed.), On educational inclusion: Meanings, history, issues and international perspectives (pp. 216-248). Routledge.

Boyle, C., Anderson, J., Page, A., & Mavropoulou, S. (Eds.) (2020). Inclusive education: Global issues & controversies (Vol. 45 in Studies in inclusive education). Boston: Brill Sense.

Kauffman, J. M., Felder, M., Ahrbeck, B., Badar, J., & Schneiders, K. (2018). Inclusion of all students in general education? International appeal for a more temperate approach to inclusion, Journal of International Special Needs Education, 21, 1-10.

I think the manuscript would also be improved by more explicit recognition of the wide range of the nature and severity of the disabilities that are involved in a blanket policy of inclusion and the differences between inclusion of all versus the inclusion of many or most. The authors do recognize that inclusion is not always possible or desirable, but the difference between encouraging and demanding inclusion needs to be discussed further, examples might be given, and alternatives discussed.

Author Response

Dear Reviewer,

We are grateful again for your review. We consider that your proposals have considerably improved our work.

From your suggestions and comments, in the Introduction section, paragraphs 2 and 3 have been deleted and we have added six new paragraphs in order to: a) explain the concept of inclusion according to the United Nations’ CRPD (referencing some suggested papers); b) explain the way the Spanish educational legislation has introduced the international perspectives of inclusion in the educational system; and c) delimit briefly some elements around the inclusion debate that may influence teachers’ attitudes.

Below we expose the changes included in the manuscript (they are also marked on the document using the “Track Changes” function).

  1. The very concept of inclusion is not well discussed (e.g., does it mean actively engaged in learning appropriate skills or does it refer to bodily presence?). Furthermore, I feel the manuscript would be greatly improved by acknowledging the ways in which inclusion in Spain fits into international perspectives, including the United Nations' CRPD and other nations' initiatives.

Changes

The concept of inclusion has been addressed during the introduction, especially in paragraph 2 on page 1. Moreover, the description of how this concept materializes in the Spanish educational system has been included in paragraphs 1-4 on page 2.

  1. I think the manuscript would also be improved by more explicit recognition of the wide range of the nature and severity of the disabilities that are involved in a blanket policy of inclusion and the differences between inclusion of all versus the inclusion of many or most.

Changes

This idea has been reflected in the changes made to the introduction, especially in paragraphs 1 and 2 on page 2.

  1. The authors do recognize that inclusion is not always possible or desirable, but the difference between encouraging and demanding inclusion needs to be discussed further, examples might be given, and alternatives discussed.

Changes

The debate about demanding and encouraging inclusive education has been set out in paragraphs 2 and 3 on page 1.

Related to this, paragraph 2 on page 13 includes the contradiction posed by an educational system in which policies tend towards more inclusive models, while teacher attitudes continue to be anchored in integrative models.

  1. Some of the changes introduced in the text (such as adding new references) have modified the order of appearance of the references in the text. Consequently, we have changed the number of most of the references in the text and in the final reference list. We have also included some minor formal changes.

Round 3

Reviewer 2 Report

I find this revision of the paper much improved. In my opinion, the authors have done a very good job of revising as suggested.